# Expression and Secretion of Intraocular Fatty Acid-Binding Protein 4 (ioFABP4) and 5 (ioFABP5) Are Regulated by Glucose Levels and Fatty Acids

**DOI:** 10.3390/ijms26051791

**Published:** 2025-02-20

**Authors:** Hiroshi Ohguro, Megumi Higashide, Erika Ishiwata, Fumihito Hikage, Megumi Watanabe, Nami Nishikiori, Tatsuya Sato, Masato Furuhashi

**Affiliations:** 1Departments of Ophthalmology, Sapporo Medical University, S1W17, Chuo-ku, Sapporo 060-8556, Japan; megumi.h@sapmed.ac.jp (M.H.); fuhika@gmail.com (F.H.); watanabe@sapmed.ac.jp (M.W.); nami076@yahoo.co.jp (N.N.); 2Departments of Cardiovascular, Renal and Metabolic Medicine, Sapporo Medical University, S1W17, Chuo-ku, Sapporo 060-8556, Japan; kurage.naritai@gmail.com (E.I.); sato.tatsuya@sapmed.ac.jp (T.S.); 3Departments of Cellular Physiology and Signal Transduction, Sapporo Medical University, S1W17, Chuo-ku, Sapporo 060-8556, Japan

**Keywords:** fatty-binding protein 4 (FABP4), fatty acid-binding protein 5 (FABP5), vascular endothelial growth factor A (VEGFA), retinal vascular disease (RVD), proliferative diabetic retinopathy (PDR), retinal vein occlusion (RVO)

## Abstract

Intraocularly, fatty acid-binding protein 4 (FABP4) and 5 (FABP5) mainly originate from human ocular choroidal fibroblasts (HOCF), and human nonpigmented ciliary epithelium (HNPCE) cells have been suggested to be pivotally involved in intraocular pathophysiology. To elucidate the unidentified regulatory mechanisms of the gene expression and protein secretion of FABPs, the effects of glucose levels, fatty acids (FAs), and peroxisome proliferator-activated receptor (PPAR) modulators were studied. To elucidate the additional biological role of FABPs, laser choroidal neovascularization (CNV) in *Fabp4^−/−^* and *Fabp4/5^−/−^* mice was analyzed by fluorescein angiography. By changing glucose levels, the secretion and expression of FABP4 in HOCF were significantly upregulated, whereas the secretion and expression of FABP5 in HNPCE decreased. The administration of various FAs, particularly docosahexaenoic acid (DHA), markedly increased the expression and secretion of both FABPs. PPAR modulators also influenced the secretion and expression of FABPs. In vivo, wild-type retina exhibited evident CNV with high fluorescein intensity, while *Fabp4^−/−^* retina showed reduced CNV formation and *Fabp4/5^−/−^* retina displayed evident CNV along with vitreous leakage. These findings suggest that (1) the production and secretion of intraocular FABP4 and FABP5 are distinctly regulated by glucose levels, FAs, and PPARs; and (2) intraocular FABP4 and FABP5 are critical for inducing retinal neovascularization and maintaining the blood-aqueous barrier.

## 1. Introduction

The family of fatty acid–binding proteins (FABPs), which have a conserved structure composed of approximately 15-kDa cytoplasmic proteins [1], and which transport fatty acids (FAs) to intracellular compartments [2,3,4,5], thereby play various physiological roles as well as roles in pathogenesis of various systemic diseases [3,4,5]. Among FABPs, FABP4 and FABP5 have been shown to be secreted into bodily fluids such as plasma [3,6,7], and the levels of these FABPs have also been shown to be significantly correlated with various pathological aspects of metabolic syndrome and cardiovascular disease [8], suggesting that FABP4 and FABP5 may be critical biomarkers for these diseases. In the ophthalmology field, our recent studies showed that both FABP4 and FABP5 are secreted into vitreous fluid and that their levels were significantly and differently correlated with retinal vascular diseases (RVDs) such as diabetic retinopathy (DR) and retinal vein occlusion (RVO), which are known to be caused by diabetes (DM) and hypertension (HT), respectively [9,10,11,12]. To determine the possible intraocular origins of intraocular FABP4 (ioFABP4) and intraocular FABP5 (ioFABP5), qPCR analysis was performed in previous studies using four different intraocularly originated cell types, including human nonpigmented ciliary epithelium (HNPCE) cells, retinoblastoma cells, adult retinal pigment epithelium 19 (ARPE19) cells, and human ocular choroidal fibroblasts (HOCF), and it was shown that the major intraocular origins of ioFABP4 and ioFABP5 were HOCF and HNPCE cells, respectively [13,14]. Furthermore, pharmacological inhibition of FABP4 in HOCF and FABP5 in HNPCE cells by BMS309403 and MF6, respectively, caused significant reductions of cellular metabolic functions [13,14]. The results of those studies suggested that both ioFABP4 and ioFABP5 may be essential factors for the maintenance of intraocular homeostasis in addition to intraocular pathogenesis. If this speculation is correct, the levels of expression and secretion of ioFABP4 and ioFABP5 may be fluctuated by several factors involved in RVD pathogenesis, such as different glucose levels, and the presence of FAs and peroxisome proliferator-activated receptor α (PPARα) and PPARγ, which have been shown to be involved in the regulation of intraocular homeostasis [15].

Therefore, in the present study, to test our speculation, the gene expression and secretion of FABP4 in HOCF and FABP5 in HNPCE cells were determined under different glucose conditions, and by administering various FAs and specific modulators of PPARα and PPARγ. In addition, to obtain additional insight into the biological roles of ioFABP4 and ioFABP5, a morphological study of laser-induced choroidal neovascularization (CNV) models was performed using FABP4-deficiency and FABP4/5-deficiency mice.

## 2. Results

The main purpose of the current study was to elucidate unidentified mechanisms by which ioFABP4 and ioFABP5 levels are regulated. As a working hypothesis, it was speculated that (1) intraocular glucose levels, (2) intraocular FAs (ioFAs), and (3) PPARs may be related based on the fact that (1) levels of ioFABP4 were positively correlated with PDR [16]; (2) levels of ioFABP4 and ioFABP5 were correlated with levels of ioFAs [12]; and (3) PPARs are known to functionally correlate with FABP4 [17,18].

### 2.1. Effects of Changing Glucose Concentrations on Expression and Secretion of ioFABP4 and ioFABP5

Initially, mRNA expression of FABP4 in HOCF and mRNA expression of FABP5 in HNPCE cells were determined by changing glucose concentrations. As shown in Figure 1, mRNA expression of FABP4 in HOCF was significantly upregulated by decreasing the glucose concentration from 25 mM to 5 mM, or by increasing the glucose concentration from 25 mM to 50 mM. In contrast, mRNA expression of FABP5 in HNPCE cells was markedly downregulated by changing glucose concentrations. The effects of changing glucose concentrations on the secretion of FABP4 from HOCF and FABP5 from HNPCE cells were determined by ELISA using the cell culture medium (Figure 2). The levels of FABP4 in the culture medium of HOCF were relatively increased and substantially increased by decreasing and increasing the glucose concentration, respectively, and FABP5 levels in the culture medium of HNPCE cells were markedly increased by changing glucose concentrations. These results suggested that the expression and secretion of ioFABP4 or ioFABP5 are increased or decreased in response to changing glucose concentrations.

### 2.2. Effects of Various FAs on Expression and Secretion of ioFABP4 and ioFABP5

Next, to study the effects of various FAs on the levels of expression and secretion of ioFABP4 and ioFABP5, HOCF and HNPCE cells were subjected to qPCR analysis and their culture media were subjected to ELISA for FABP4 and FABP5. As shown in Figure 3 and Figure 4, among the 5 FAs, (1) administration of DHA significantly increased the mRNA expression levels of FABP4 in HOCF, and Ara and DHA also increased levels of FABP4 in the culture medium; and (2) EPA and DHA induced marked increases in the mRNA expression levels of FABP5 in HNPCE cells, and Ara, EPA and DHA significantly increased levels of FABP4 in the culture medium. These results suggested that the expression and secretion of ioFABP4 or ioFABP5 are significantly modulated by FAs, especially DHA.

### 2.3. Effects of PPARs on Expression and Secretion of ioFABP4 and ioFABP5

The contribution of the lipid metabolism regulators PPARα and PPARγ to the gene expression and secretion of ioFABP4 and ioFABP5 was studied by using their specific agonists and antagonists. As shown in Figure 5 and Figure 6, (1) the mRNA expression of FABP4 in HOCF was decreased by agonists for PPARα and PPARγ and increased by the antagonist for PPARγ, respectively; and (2) the mRNA expression of FABP5 in HNPCE cells was decreased by the agonist for PPARα. In contrast, significant changes in the secretion of FABP4 from HOCF and FABP5 from HNPCE cells were not observed, except for a substantial decrease in FABP5 in HNPCE cells by the PPARγ antagonist. The results indicated that the expression and secretion of ioFABP4 and ioFABP5 may also be affected by modulators for PPARs.

### 2.4. Possible Effects of ioFABP4 and ioFABP5 on Laser-Induced CNV

Next, to elucidate unidentified pathophysiological roles of ioFABP4 and ioFABP5, an in vivo mouse model of laser-induced CNV that is commonly used for studying the pathogenesis of RVDs [19] was assessed using *Fabp4^−/−^* and *Fabp4/5^−/−^* mice in addition to WT mice, and an analysis was performed by fluorescein angiography. As shown in Figure 7, the appearance of CNV in WT mice and in *Fabp4^−/−^* mice was significantly different. A high fluorescein intensity area (*) corresponding to laser spots was observed in the WT mouse retina, while in the *fabp4^−/−^* mouse retina, high fluorescein intensity was observed around laser spots. However, fluorescein intensity at the laser spots was low (#), suggesting that CNV formation after a laser burn may be suppressed in the *fabp4^−/−^* mouse retina compared with that in the WT mouse retina. To support this speculation, upon laser exposure, the mRNA expression of vegf was significantly upregulated in WT mouse retina, but was less in *fabp4^−/−^* mouse retina. In turn, inflammatory cytokines *il6* and *il1β*, chemokine, *ccl2*, and *tnf* of *fabp4^−/−^* mouse retina were significantly upregulated by laser exposure compared to WT mouse retina (Figure 8). More interestingly, in the *Fabp4/5^−/−^* mouse retina, vitreous leakage of fluorescein dye was observed in addition to the characteristic appearance of CNV observed in the *fabp4^−/−^* mouse retina. These results suggested that (1) ioFABP4 may be a crucial factor for inducing retinal neovascularization; and (2) ioFABP5 may be significantly involved in the blood–ocular barrier, presumably the blood–aqueous barrier, since the intraocular origin of ioFABP5 is HNPCE cells.

## 3. Discussion

In contrast to the mRNA expression of ioFABP4 detected mostly in HOCF in the intraocular environment, our previous study showed that there was positive immunostaining against FABP4 over all of the retinal segments, except the photoreceptor outer segment (OS) [12], suggesting that ioFABP4 secreted from the ocular choroid may spread into the sensory retina beyond the RPE cell layer. In support of this speculation, amplitudes of the a-wave and b-wave of an electroretinogram were significantly enhanced in the *Fabp4^−/−^* mouse retina compared to the WT mouse retina [12]. In addition, it has been postulated that the retinal choroid and RPE form an “RPE/choroid complex” that may play cooperative roles in retinal pathophysiology [20,21], since retinal choroid in addition to the RPE is involved in various intraocular diseases such as age-related macular degeneration (AMD) [22] and myopia [23]. In fact, real-time cellular metabolic analysis using a Seahorse bioanalyzer showed that pharmacological inhibition of FABP4 in HOCF by BMS309403 led to pseudohypoxic changes that may be a critical cause of neovascularization in AMD [24] and malignant tumors [25,26]. If this is the case, the results of the present study, showing that the expression and secretion of FABP4 in HOCF were increased by changing glucose concentrations, may rationally explain the currently unidentified clinical issue related to DR, that is, why worsening of DR is caused by rapid improvement of blood glucose levels [27]. In a previous study, the abnormally upregulated expression of FABP4 was observed in retinopathy, and FABP4 deficiency thus improves pathological retinal vascularization, suggesting a possible protection against retinopathy [28]. The present study also showed that the formation of laser-induced CNV was remarkably suppressed in the *Fabp4^−/−^* mouse retina.

In contrast to FABP4 in HOCF, we found that the secretion, but not expression, of FABP5 in HNPCE cells was significantly decreased in response to changes in glucose levels. Our previous study suggested that ioFABP5 may be involved in the regulation of aqueous humor production in HNPCE cells due to the maintenance of an unfolded protein response (UPR) and an aquaporin1 (AQP1)-related mechanism, thereby regulating levels of intraocular pressure (IOP) [14]. Furthermore, the present study showed that there was vitreous leakage of fluorescein dye in the *Fabp4/5^−/−^* mouse retina in addition to the characteristic changes in the appearance of CNV in the *Fabp4^−/−^* mouse retina, suggesting that FABP5 in HNPCE cells may be critically involved in the regulation of aqueous humor production and secretion. Furthermore, since the mRNA expression of various inflammatory-related factors in the eye cup obtained from the *fabp4/5^−/−^* mouse did not reach detectable levels, we speculated that the intraocular levels of these inflammatory-related factors may be diluted through linkage with the peripheral blood circulation, due to deteriorated function of blood–aqueous barrier. Taken together with the consensus that FABP5 plays significant roles in several aspects of metabolic syndrome, including hypertension and atherosclerosis [3,7], there may be some FABP5-dependent interplay between levels of IOP and hypertension. In fact, our recent study showed a significant correlation between levels of IOP and a new onset of hypertension during a 10-year follow-up period in 7,487 Japanese subjects who underwent medical health checkups in 2006 [29], indicating that a high level of IOP, even within the normal range, is an independent risk factor for the development of hypertension over a 10-year period [30]. As of this writing, although we do not know why the fluctuation in the expression and secretion of FABP4 in HOCF and that of FABP5 in HNPCE cells induced by changes in glucose levels were different, the effects of the fluctuation of osmotic pressure in response to changes in glucose concentrations on the secretion of FABP5 may be related. In fact, it has been shown that osmotic pressure is one of the critical factors for the maintenance of homeostasis of aqueous humor production [31,32]. Furthermore, a recent observation that the expression of AQP1 and AQP2 and aqueous humor osmolality were significantly altered in a glaucomatous eye rationally supports our idea, because the expression of AQP1 in HNPCE cells was also significantly modulated by FABP5 [14].

Regarding the effects of FAs on the production and secretion of FABP4, it was shown that omega-3 fatty acids decrease the serum FABP4 level, possibly by reducing the expression and consecutive secretion of FABP4 in adipocytes [6], and that FABP4 gene expression was positively correlated (*p* < 0.05) with EPA and DHA in lamb muscle [33], suggesting that the effects of FAs on FABP4 may be different among various cell types. In the present study, the expression and secretion of both FABP4 in HOCF and FABP5 in HNPCE cells were significantly increased by FAs, especially omega-3 fatty acids, EPA, and DHA. Interestingly, previous studies showed that FABP5 binding is significantly involved in the uptake of DHA in brain endothelial cells and the subsequent transport of DHA to the blood–brain barrier (BBB) [34], and that dietary DHA supplementation enhances the expression of FABP 5 at the BBB and brain DHA levels [35], suggesting that FABP5 may be pivotally involved in the homeostasis of the BBB. Similarly, increases in the expression and secretion of FABP4 and FABP5 were induced by the omega-3 fatty acids EPA and DHA, and were observed in HOCF and HNPCE cells, respectively. HOCF and HNPCE cells are known to be critical cells constituting the outer blood–retinal barrier (oBRB) [36] and blood–aqueous barrier (BAB) [37,38], respectively. Collectively, the results suggest that regulation of the expression and secretion of FABP4 and FABP5 is extremely important for maintaining the biological barrier of the brain and intraocular environment.

Based on the facts that the expression of PPARγ is inhibited in the retina in a high-glucose environment, and that the PPARγ agonist rosiglitazone thus induces a delay in the onset and progression of DR [39], it has been suggested that activation of PPARγ may be an effective strategy for improving DR [40]. Regarding FABP4 and PPARγ, it was shown that the FABP4 gene contains a peroxisome proliferation response element [41], and that FABP4 can directly interact with PPARγ, thereby inducing the ubiquitination and subsequent degradation of PPARγ [17]. In addition, a recent study has shown that PPARγ was inactivated in diabetic mice and in high glucose-induced ARPE-19 cells [42]. It was also reported that the activity of PPARγ was inhibited by FABP4, which is consistent with our observation that FABP4 was a factor that deteriorates DR pathogenesis [13]. Alternatively, regarding PPARα, its selective agonist pemafibrate was reported to up-regulate expression levels of downstream PPARα targets, including acyl-CoA oxidase 1, FABP4, and fibroblast growth factor 21 in the liver, but not in the retina [43]. However, in the present study, the expression and secretion of FABP4 in HOCF and FABP5 in HNPCE cells were not significantly altered. Therefore, it was thought that the contribution of PPARs may not be important for regulation of the expression and secretion of ioFABP4 and ioFABP5.

As limitations of this study, the following issues need to be investigated. Firstly, the biological aspects of the commercially available HOCF and HNPCE cells may not be identical to in vivo native and matured cells. In fact, in contrast to the present result that PPARα activation by Pema induced significant downregulation of FABP4 in HOCF, a recent in vivo study has shown that Pema-induced effects on the expression of FABP4 was exclusively organ and tissue dependent; that is, Pema induced a significant increase of *fabp4* in liver but not retina and RPE-choroid in mice, and those effects were enhanced during increasing exposure periods [44]. Therefore, the effects of the modulation of PPARs may be variable in various experimental conditions, such as in vitro and in vivo, different time points, and different doses. Secondly, the laser CNV model may not replicate the exact molecular mechanism of pathogenesis of human RVDs. Thirdly, the reason why the expression profile of FABP4 and FABP5 was different even in intraocularly originated cells remains to be elucidated. Fourthly, the regulatory mechanism for the expression and secretion of ioFABP4 and ioFABP5 has not been fully understood. In fact, as shown in Figure 1A and Figure 2A, the mRNA expression of FABP4 was inconsistent with the secretion of FABP4 in HOCF, suggesting additional underlying mechanisms to control the secretion of intracellularly expressed FABP4 in HOCF. Fifthly, our idea that ioFABP4 and ioFABP5 may play pivotal roles in inducing retinal neovascularization and maintaining the blood–aqueous barrier is still speculative. Therefore, additional investigations for revealing unidentified mechanisms using in vitro additional functional assays, and in vivo experiments using a mice model with RVDs, will be required.

## 4. Materials and Methods

### 4.1. Preparations of HOCF and HNPCE Cells

All experimental procedures using commercially available human-derived cells were conducted after approval by the internal review board of Sapporo Medical University. Human ocular choroidal fibroblasts (HOCF, Cat. #6620, Science Research Laboratories, Inc., Carlsbad, CA, USA) and HNPCE cells (Cat. #6580, Science Research Laboratories, Inc., Carlsbad, CA, USA) were purchased and separately cultured in 150-mm planar culture dishes until they reached 90% confluence at 37 °C in a growth medium composed of high-glucose (25 mM glucose) DMEM containing 10% FBS, 1% L-glutamine, and 1% antibiotic-antimycotic. These cells were maintained by changing the medium every other day under standard humid normoxia conditions (37 °C, 5% CO_2_).

For changing glucose concentrations (1) from 25 mM to 5.5 mM, or (2) from 25 mM to 50 mM, (1) the growth medium containing 25 mM glucose, as described above, was changed to low-glucose (5.5 mM glucose) DMEM containing 10% FBS, 1% L-glutamine, and 1% antibiotic-antimycotic; or (2) to adjust the glucose concentration at 50 mM, a suitable volume of 50% glucose was added to the growth medium containing 25 mM glucose, as described above, and the cells were further cultured for 24 h.

To study the effects of various FAs, cells were incubated for 24 h with 100 μM of C16:0 palmitic acid (Pal); C18:1 oleic acid (Ole); C22:4 arachidonic acid (Ara); C20:5 eicosapentaenoic acid (EPA); or C22:6 docosahexaenoic acid (DHA) after changing the culture medium in which FBS was substituted by fatty acid-free albumin for 24 h. The concentrations of FAs were determined according to those detected in vitreous specimens obtained from patients with epiretinal membrane or retinal vascular diseases in our previous studies [12,45].

Based on previous studies [46,47,48,49,50,51], the effects of PPARα and PPARγ on cells were examined by incubation for 24 h with the PPARα agonist (10 μM pemafibrate [46,47]), the PPARα antagonist (20 μM GW7647 [48]), the PPARγ agonist (10 μM rosiglitazone [49,50]), and the PPARγ antagonist (0.1 μM T0070907 [51]).

### 4.2. Laser CNV Model

Wild-type (WT), *fabp 4^−/−^* and *fabp 4 and 5^−/−^* male mice aged seven weeks were generated in the laboratory of Dr. Gökhan S. Hotamisligil (Harvard T.H. Chan School of Public Health) and backcrossed for more than eight generations into a C56BL/6J genetic background. All of the animals were reared under conditions of free access to food and water as well as cyclic light conditions (12 h on/12 h off) until the experiments were initiated. As reported previously [19,52], laser photocoagulation was used to induce CNV in mice. After dilating pupils, mice were placed on a platform under a slit lamp, and the corneas were anesthetized with oxybuprocaine hydrochloride eye drops (Santen Pharmaceutical Co., LTD., Osaka, Japan). Then, 532 nm argon laser-induced photocoagulation was used to disrupt Bruch’s membrane bilaterally in each mouse. Three to four spots of laser photocoagulation in the posterior pole of the retina were created with a power of 150–200 mW, duration of 100 ms and spot size of 50 μm using a slit lamp delivery system (slit lamp: model SL1000, laser: purepoint, ALCON Inc. Geneva, Switzerland) with a handheld coverslip as a contact lens. The laser spots were located approximately two to three optic disc diameters away from the optic nerve head, avoiding the main vessels. The appearance of a white bubble, which indicates a break in Bruch’s membrane, is an important factor for obtaining CNV, and only burns causing bubble formation were included in subsequent experiments. Spots with hemorrhage or failing to develop a bubble were excluded from the analysis.

### 4.3. Fluorescein Angiography (FAG)

FAG was performed after intraperitoneal injection of 1 mL of 10% fluorescein sodium using the fundus camera (model Vx-10α, Kowa Co., Ltd., Tokyo, Japan) at a 50° angle of view. Images were recorded 3 to 240 s after injections.

### 4.4. Gene Expression Analysis

By using planar cultured cells and an eye cup obtained from WT and *fabp*-deficiency mice through the removal of the cornea and lens, real-time PCR was carried out essentially as previously reported [53] using predesigned primers (Appendix A). The expression of each gene was normalized using the expression of the housekeeping gene 36B4 (Rplp0).

### 4.5. Other Analytical Methods

The concentrations of FABP4 and FABP5 in cell culture media were measured using commercially available enzyme-linked immunosorbent assay kits for FABP4 (Biovendor R&D, Modrice, Czech Republic) and FABP5 (Biovendor R&D, Modrice, Czech Republic). As experimental data, the arithmetic mean ± the standard error of the mean (SEM) was used in conjugation with statistical analyses, essentially as described in our previous report [53].

## 5. Conclusions

The present results highlight that the production and secretion of ioFABP4 and ioFABP5 are differently regulated by different glucose levels, FAs, and PPARs, and suggested that ioFABP4 may contribute to retinal neovascularization, while ioFABP5 may be involved in the maintenance of the aqueous–blood barrier through regulation of aqueous humor production. These results suggest that both ioFABPs play pivotal roles in homeostasis as well as pathogenesis in the intraocular environment, and future investigations for revealing additional unidentified biological significance of ioFABPs using additional in vitro and in vivo experiments will be necessary.

## Figures and Tables

**Figure 1 ijms-26-01791-f001:**
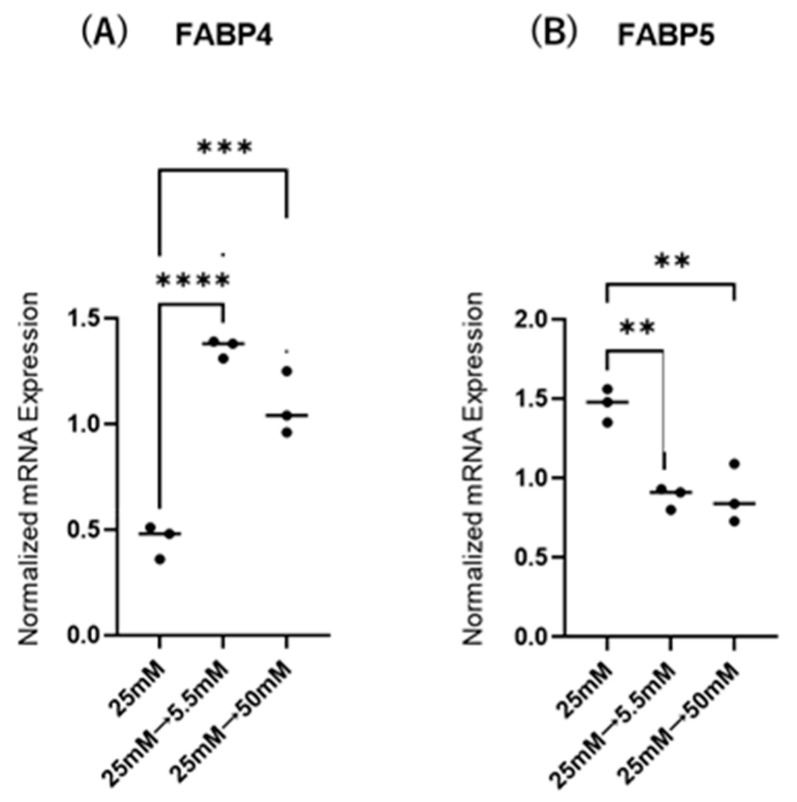
Effects of different glucose conditions on mRNA expression of *FABP4* in HOCF and *FABP5* in HNPCE cells. Planar cultured HOCF (**A**) and HNPCE cells (**B**) were prepared under a low glucose (5.5 mM) condition for 5 days. The cells were further cultured for 24 h (control) or were further cultured under a high glucose (50 mM) condition for 24 h. Each specimen was subjected to qPCR analysis and the mRNA expression levels of *FABP4* and *FABP5* were evaluated. Duplicated experiments were performed using fresh preparations (n = 5 each). ** *p* < 0.01, *** *p* < 0.005, **** *p* < 0.001.

**Figure 2 ijms-26-01791-f002:**
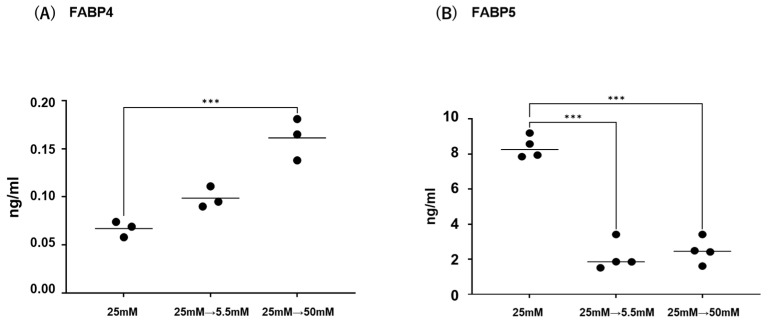
Effects of different glucose conditions on the secretion of *FABP4* in HOCF and *FABP5* in HNPCE cells. Planar cultured HOCF (**A**) and HNPCE cells (**B**) were prepared under a low glucose (5.5 mM) condition for 5 days. The cells were further cultured for 24 h (control) or were further cultured under a high glucose (50 mM) condition for 24 h. Each culture medium was collected and subjected to ELISA for *FABP4* and *FABP5*. Duplicated experiments were performed using fresh preparations. (n = 3 each). *** *p* < 0.005.

**Figure 3 ijms-26-01791-f003:**
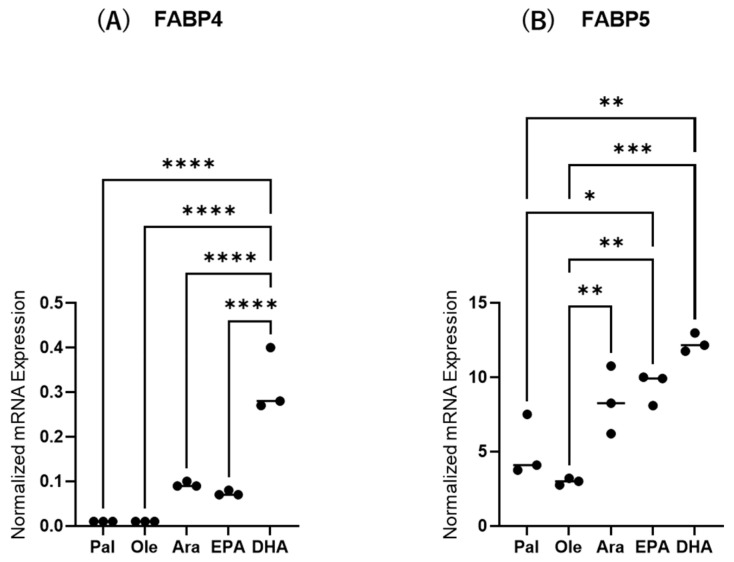
Effects of various free fatty acids on the mRNA expression of *FABP4* in HOCF and *FABP5* in HNPCE cells. Planar cultured HOCF (**A**) and HNPCE cells (**B**) were prepared under a 25 mM glucose condition for 5 days. After changing the culture medium in which FBS was substituted by fatty acid-free albumin for 24 h, the cells were further cultured for 24 h by administering 100 μM each of various free fatty acids including Pal, Ole, Ara, EPA and DHA. Each specimen was subjected to qPCR analysis and the mRNA expression levels of *FABP4* and *FABP5* were evaluated. Duplicated experiments were performed using fresh preparations (n = 5 each). * *p* < 0.05, ** *p* < 0.01, *** *p* < 0.005, **** *p* < 0.001.

**Figure 4 ijms-26-01791-f004:**
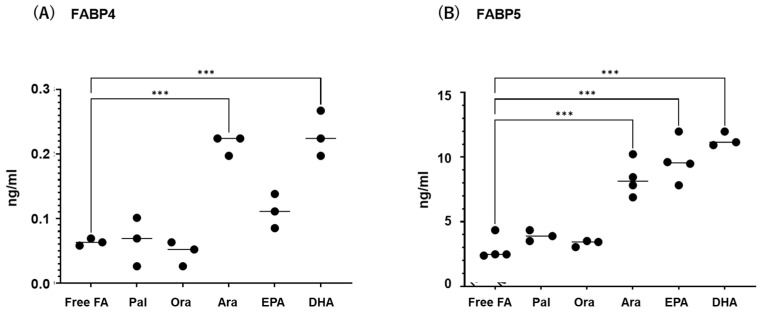
Effects of various free fatty acids on the secretion of *FABP4* in HOCF and *FABP5* in HNPCE cells. Planar cultured HOCF (**A**) and HNPCE cells (**B**) were prepared under a 25 mM glucose condition for 5 days. After changing the culture medium in which FBS was substituted by fatty acid-free albumin for 24 h, the cells were further cultured for 24 h by administering 100 μM each of various free fatty acids including Pal, Ole, Ara, EPA and DHA. Each culture medium was collected and subjected to FLISA for *FABP4* and *FABP5* were evaluated. Duplicated experiments were performed using fresh preparations (n = 3 each). *** *p* < 0.005.

**Figure 5 ijms-26-01791-f005:**
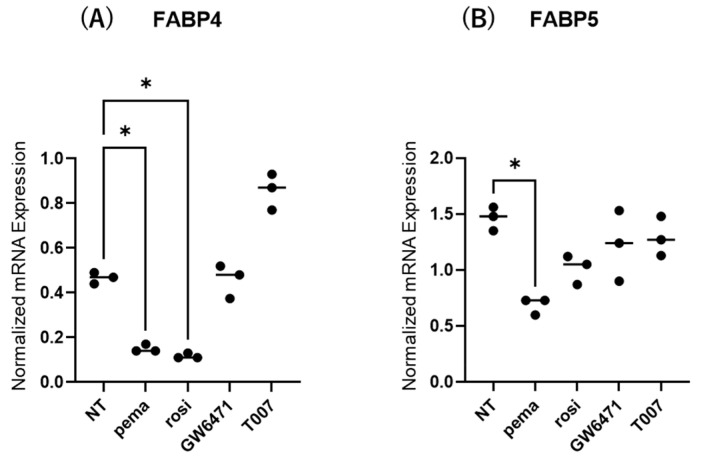
Effects of modulators for PPARa and PPARg on the mRNA expression of *FABP4* in HOCF and HNPCE cells. Planar cultured HOCF (**A**) and HNPCE cells (**B**) were prepared under a 25 mM glucose condition for 5 days. The cells were further cultured for 24 h by administering the PPARα agonist (pema: 10 μM pemafibrate), the PPARγ agonist (rosi: 10 μM rosiglitazone), the PPARα antagonist (GW6471: 20 μM GW6471) and the PPARγ antagonist (T007: 0.1 μM T0070907). Each specimen was subjected to qPCR analysis and the mRNA expression levels of *FABP4* and *FABP5* were evaluated. Duplicated experiments were performed using fresh preparations (n = 5 each). * *p* < 0.05. NT: non-treated control.

**Figure 6 ijms-26-01791-f006:**
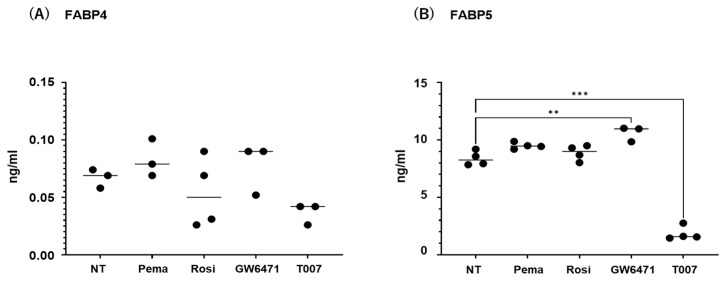
Effects of modulators for PPARa and PPARg on the secretion of *FABP4* in HOCF and HNPCE cells. Planar cultured HOCF cells (**A**) and HNPCE cells (**B**) were prepared under a 25 mM glucose condition for 5 days. The cells were further cultured for 24 hrs by administering the PPARα agonist (pema: 10 μM pemafibrate), the PPARγ agonist (rosi: 10 μM rosiglitazone), the PPARα antagonist (GW6471: 20 μM GW6471) and the PPARγ antagonist (T007: 0.1 μM T0070907). Each culture medium was collected and subjected to FLISA for *FABP4* and *FABP5* were evaluated. Duplicated experiments were performed using fresh preparations (n = 3 each). ** *p* < 0.01. NT: non-treated control. *** *p* < 0.005.

**Figure 7 ijms-26-01791-f007:**
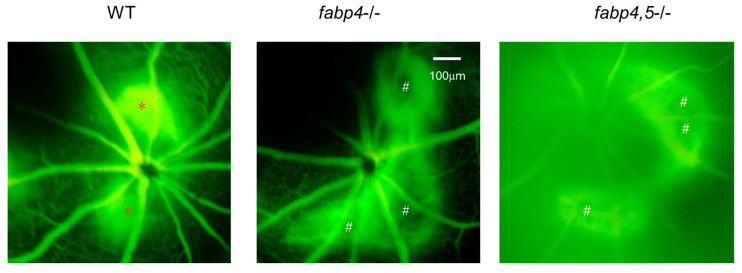
Representative photos of fluorescein retinal angiography of laser-induced choroidal neovascular models using wild-type, *fabp4^−/−^* and *fabp4/5^−/−^* mice. After intraperitoneal injection of 1 mL of 10% fluorescein sodium, late-phase of fluorescein angiography (90–180 s after injection) of mice (n = 5 each) induced by laser-CNV was observed by a fundus camera as described in Methods. Representative images are shown. *: lesion of active CNV, #: CNV not produced-lesion.

**Figure 8 ijms-26-01791-f008:**
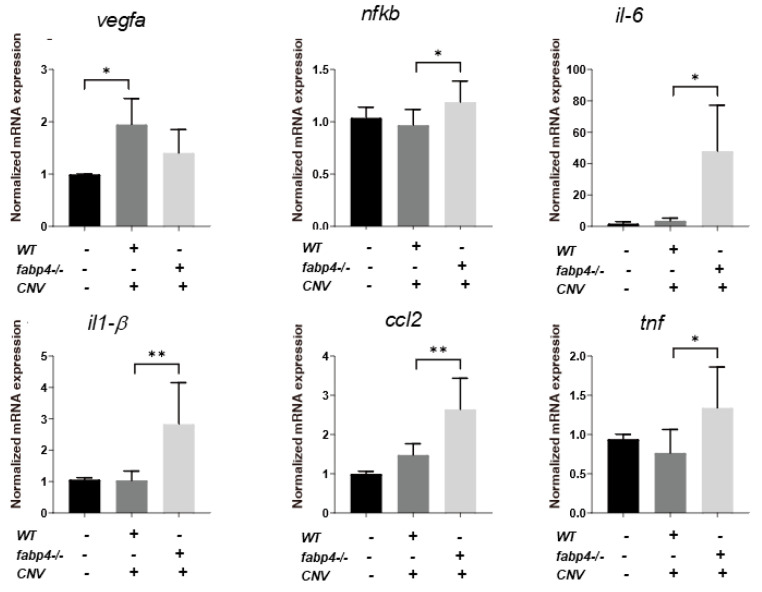
Gene expression of *vegfa*, *nfkb*, *il-6*, *il-1β*, *ccl2* and *tnf* of laser-induced choroidal neovascular models using wild type and *fabp4^−/−^* mice. An eye cup obtained from wild-type (WT), *fabp4^−/−^* (*fabp*) mice that were untreated or treated by laser-CNV induction was subjected to qPCR analysis, and the levels of mRNA expression of *vegfa*, *nfkb*, *il-6*, *il-1β*, *ccl2*, and *tnf* were plotted. Duplicated experiments were performed using fresh preparations (n = 5 each). * *p* < 0.05, ** *p* < 0.01. vegfa: vascular endothelial growth factor A, nfkb: nuclear factor kappa B, il-6: interleukin-6, il1-β: interlekin 1-β, ccl2: C-C motif chemokine ligand 2, tnf: tumor necrosis factor.

## Data Availability

The original contributions presented in this study are included in the article/Appendix A; further inquiries can be directed to the corresponding author.

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
