# Peer review of "Expression and Secretion of Intraocular Fatty Acid-Binding Protein 4 (ioFABP4) and 5 (ioFABP5) Are Regulated by Glucose Levels and Fatty Acids"

_ijms, 2025, doi:10.3390/ijms26051791_

Round 1
Reviewer 1 Report
Comments and Suggestions for Authors
The manuscript is interesting. Several questions and comments are listed.
FABPs' roles and expressions are basically examined in vitro. Based on Figure 7, it seems that in vivo work is available. However, their expressions and deep roles in vivo are not well examined. In vitro's limitation should be discussed. And if possible, adding in vivo outcomes could be valuable.
For PPAR alpha activation, it has been reported to change the expression of ocular FABP4. Screening data from the liver, retina, and RPE-Choroid showed that FABP4 expression could be increased by a PPAR alpha activator (DOI10.7717/peerj.14611). There is a discrepancy in vitro and in vivo. Depending on the time point and the dose of the activator, the trend could be changed in vitro. This should be at least discussed.
For in vivo's CNV data, the characterization of the model is not well done and the condition of the KO should be well presented with biomolecular assays. Also, the outcomes' reasons should be experimentally evidenced.
Each fatty acid should be treated with optimal doses. The current data do not contain this aspect. Depending on the dose and time point, the in vitro dat could be changed. Its aspect should be cautious.
Author Response
Dear Editor,
Thank you very much for the constructive comments concerning our manuscript “Expression and secretion of intraocular fatty acid-binding protein 4 (ioFABP4) and ioFABP5 are regulated by glucose levels and fatty acids.”. We carefully checked all of the reviewers’ comments and prepared a revised version of our paper that takes these comments into account. The changes are listed below.
Reviewer 1 comments
The manuscript is interesting. Several questions and comments are listed.
FABPs' roles and expressions are basically examined in vitro. Based on Figure 7, it seems that in vivo work is available. However, their expressions and deep roles in vivo are not well examined. In vitro's limitation should be discussed. And if possible, adding in vivo outcomes could be valuable.
- For PPAR alpha activation, it has been reported to change the expression of ocular FABP4. Screening data from the liver, retina, and RPE-Choroid showed that FABP4 expression could be increased by a PPAR alpha activator (DOI10.7717/peerj.14611). There is a discrepancy in vitro and in vivo. Depending on the time point and the dose of the activator, the trend could be changed in vitro. This should be at least discussed.
Answer; We sincerely appreciate your excellent comment. As pointed out, in the results of suggested study by Lee et al. (DOI10.7717/peerj.14611), Pema induced significant increase of Fabp4 in liver but not retina and RPE-choroid. Therefore, I totally agree that effects of PPARs modulators could be different depending on various experimental conditions such as in vivo vs in vitro, different time points, the dose of modulators in addition to different organs and tissues. Therefore, this information is included in Discussion: ‘Firstly, the biological aspects of the commercially available HOCF and HNPCE cells may not be identical to in vivo native and matured cells. In fact, in contrast to the present result that PPARa activation by Pema induced significant downregulation of FABP4 in HOCF, a recent in vivo study has shown that Pema-induced effects on the expression of FABP4 was exclusively organ and tissue dependent, that is, Pema induced significant increase of fabp4 in liver but not retina and RPE-choroid in mice and those effects were enhanced during increasing exposure periods [44]. Therefore, effects of modulation of PPARs may be variable in various experimental conditions, such as in vitro and in vivo, different time points and different doses.’.
- For in vivo's CNV data, the characterization of the model is not well done and the condition of the KO should be well presented with biomolecular assays. Also, the outcomes' reasons should be experimentally evidenced.
Answer; We sincerely appreciate your excellent comment. I agree that our in vivo CNV is still preliminary despite that fluorescein angiography showed significant difference among WT and FABP KO mice. Therefore, as suggested, additional qPCR data (fig. 8) of WT and FABP4 KO mice eyes are included. Nevertheless, experiment of FABP4/5 KO mice eyes did not work because linkage with peripheral circulation may disturb qPCR experiment due to break down of aqueous blood barrier. In support of this, previous our study showed significant upregulation of aquaporin1 in HNPCE cells by pharmacological inhibition of FABP5 (ref#14). This information is included in Discussion: ‘Furthermore, since mRNA expression of various inflammatory-related factors in eye cup obtained from fabp4/5-/- mouse did not reach to detectable levels, we speculated that intraocular levels of those inflammatory-related factors may be diluted by linkage with peripheral blood circulation due to deteriorated function of blood aqueous barrier.’.
- Each fatty acid should be treated with optimal doses. The current data do not contain this aspect. Depending on the dose and time point, the in vitro dat could be changed. Its aspect should be cautious.
Answer; We sincerely appreciate your excellent comment. In terms of concentrations of administering FAs, we used 100 mM based on the previous our results of measurement of FAs’ concentrations in vitreous specimens obtained from patients without and with retinal vascular diseases (ref#12). I agree that time points should be very important. However, in our previous study, we showed that around 100 mM of FAs are detected in vitreous specimens obtained from several patients suggesting that FAs should permanently present. In addition, I do not think that FAs concentration fluctuated rapidly. Therefore, I think that we should focus longer exposure rather than short exposure. In addition, short exposure experiment may be important for evaluating toxic effects of FAs. Alternatively, I afraid that longer exposure more than 24hours may induce unexpected and unphysiological effects of FAs themselves and cell homeostasis.
Reviewer 2 comments
This article elucidates the unknown regulatory mechanism of FABPs expression and secretion through the effects of glucose levels, fatty acids and peroxisome proliferator activated receptor modulators. The morphology of laser-induced choroidal neovascularization model in FABP4 deficient and fabp4g5 deficient mice was studied. The results showed that FABPs played a key role in the homeostasis and pathogenesis of intraocular environment. However, there are still some defects in this article:
- all abbreviations appearing in the abstract shall be attached with their full names;
Answer; We sincerely appreciate your excellent comment. As pointed out, abstract is revised to attach full names of all abbreviations: ‘Intraocularly, fatty acid-binding protein 4 (FABP4) and FABP5 mainly originate from human ocular choroidal fibroblasts (HOCF), and human nonpigmented ciliary epithelium (HNPCE) cells have been suggested to be pivotally involved in intraocular pathophysiology. To elucidate the unidentified regulatory mechanisms of the gene expression and protein secretion of FABPs, effects of glucose levels, fatty acids (FAs), and peroxisome proliferator-activated receptor (PPAR) modulators were studied. To elucidate additional biological role of FABPs, laser choroidal neovascularization (CNV) in Fabp4-/- and Fabp4/5-/- mice was analyzed by fluorescein angiography. By changing glucose levels, secretion and expression of FABP4 in HOCF were significantly upregulated, whereas secretion and expression of FABP5 in HNPCE decreased. Administration of various FAs, particularly docosahexaenoic acid (DHA), markedly increased expression and secretion of both FABPs. PPAR modulators also influenced secretion and expression of FABPs. In vivo, wild-type retina exhibited evident CNV with high fluorescein intensity, while Fabp4-/- retina showed reduced CNV formation and Fabp4/5-/- retina displayed evident CNV along with vitreous leakage. These findings suggest that 1) production and secretion of intraocular FABP4 and FABP5 are distinctly regulated by glucose levels, FAs, and PPARs, and 2) intraocular FABP4 and FABP5 are critical for inducing retinal neovascularization and maintaining the blood-aqueous barrier.’.
- the result part is somewhat long and can be divided into several summaries;
Answer; We sincerely appreciate your excellent comment. As suggested, the result part is divided by subheading and sumarries.
- why can't FABP4 and FABP5 be detected in the same cell?
Answer; We sincerely appreciate your excellent comment. Currently, although we do not know why can't FABP4 and FABP5 be detected in the same cell, we speculate that FABP4 and FABP5 play different roles of intraocular physiology and pathogenesis. Therefore, this information is included in the study limitation in Discussion:’ As limitations of this study, the following issues need to be investigated. Firstly, the biological aspects of the commercially available HOCF and HNPCE cells may not be identical to in vivo native and matured cells. In fact, in contrast to the present result that PPAR activation by Pema induced significant downregulation of FABP4 in HOCF, a recent in vivo study has shown that Pema-induced effects on the expression of FABP4 was exclusively organ and tissue dependent, that is, Pema induced significant increase of fabp4 in liver but not retina and RPE-choroid in mice and those effects were enhanced during increasing exposure periods [44]. Therefore, effects of modulation of PPARs may be variable in various experimental conditions, such as in vitro and in vivo, different time points and different doses. Secondly, the laser CNV model may not replicate the exact molecular mechanism of pathogenesis of human RVDs. Thirdly, the reason why expression profile of FABP4 and FABP5 was different even though intraocularly originated cells remains to be elucidated. Fourthly, regulatory mechanism for expression and secretion of ioFABP4 and ioFABP5 has not been fully understood. In fact, as shown in (Fig. 1a and Fig. 2a), mRNA expression of FABP4 was inconsistent with secretion of FABP4 in HOCF, suggesting that additional underlying mechanisms to control secretion of intracellularly expressed FABP4 in HOCF. Fifthly, our idea that ioFABP4 and ioFABP5 may have pivotal roles for inducing retinal neovascularization and maintaining the blood aqueous barrier is still speculative. Therefore, additional investigations for revealing unidentified mechanisms using in vitro additional functional assays and in vivo experiments using a mice model with RVDs will be required.’.
- why is the trend of fabp4mrna expression and FABP4 secretion inconsistent in hocf cells (Fig. 1a and Fig. 2a)?
Answer; We sincerely appreciate your excellent comment. Currently, although we do not know the trend of FABP4 mRNA expression and FABP4 secretion inconsistent in HOCF cells (Fig. 1a and Fig. 2a), we speculate that gene expression in HOCF and secretion of FABP4 from HOCF may be differently regulated in response to changing glucose concentrations. Therefore, this information is included in the study limitation in Discussion:’ As limitations of this study, the following issues need to be investigated. Firstly, the biological aspects of the commercially available HOCF and HNPCE cells may not be identical to in vivo native and matured cells. In fact, in contrast to the present result that PPARa activation by Pema induced significant downregulation of FABP4 in HOCF, a recent in vivo study has shown that Pema-induced effects on the expression of FABP4 was exclusively organ and tissue dependent, that is, Pema induced significant increase of fabp4 in liver but not retina and RPE-choroid in mice and those effects were enhanced during increasing exposure periods [44]. Therefore, effects of modulation of PPARs may be variable in various experimental conditions, such as in vitro and in vivo, different time points and different doses. Secondly, the laser CNV model may not replicate the exact molecular mechanism of pathogenesis of human RVDs. Thirdly, the reason why expression profile of FABP4 and FABP5 was different even though intraocularly originated cells remains to be elucidated. Fourthly, regulatory mechanism for expression and secretion of ioFABP4 and ioFABP5 has not been fully understood. In fact, as shown in (Fig. 1a and Fig. 2a), mRNA expression of FABP4 was inconsistent with secretion of FABP4 in HOCF, suggesting that additional underlying mechanisms to control secretion of intracellularly expressed FABP4 in HOCF. Fifthly, our idea that ioFABP4 and ioFABP5 may have pivotal roles for inducing retinal neovascularization and maintaining the blood aqueous barrier is still speculative. Therefore, additional investigations for revealing unidentified mechanisms using in vitro additional functional assays and in vivo experiments using a mice model with RVDs will be required.’.
- what does nt mean in figures 5 and 6? Which are PPAR α and PPAR γ agonists and antagonists?
Answer; We sincerely appreciate your excellent comment. As pointed out, we show NT (non-treated) and which are PPAR α and PPAR γ agonists and antagonists in the legends of Figs 5 and 6.
- how to determine the concentration of PPAR α and PPAR γ agonists and antagonists?
Answer; We sincerely appreciate your excellent comment. I apologize our careless mistake of PPAR modulators by missing changing symbolic font. Therefore, mM but not mM concentrations were used and references showing dose of these modulators are suitable concentrations are included.

Reviewer 2 Report
Comments and Suggestions for Authors
This article elucidates the unknown regulatory mechanism of FABPs expression and secretion through the effects of glucose levels, fatty acids and peroxisome proliferator activated receptor modulators. The morphology of laser-induced choroidal neovascularization model in FABP4 deficient and fabp4g5 deficient mice was studied. The results showed that FABPs played a key role in the homeostasis and pathogenesis of intraocular environment. However, there are still some defects in this article:
- all abbreviations appearing in the abstract shall be attached with their full names;
- the result part is somewhat long and can be divided into several summaries;
- why can't FABP4 and FABP5 be detected in the same cell?
- why is the trend of fabp4mrna expression and FABP4 secretion inconsistent in hocf cells (Fig. 1a and Fig. 2a)?
- what does nt mean in figures 5 and 6? Which are PPAR α and PPAR γ agonists and antagonists?
- how to determine the concentration of PPAR α and PPAR γ agonists and antagonists?
Author Response

(The authors gave the same response as above.)

Round 2
Reviewer 1 Report
Comments and Suggestions for Authors
The raised concerns have been addressed.
Reviewer 2 Report
Comments and Suggestions for Authors
The author answered all my questions and made revisions in the revised manuscript. These show the author's serious attitude. At the same time, I noticed that the author pointed out the limitations of the article. I acknowledge the work done by the author.